



# Precipitation-fire linkages in Indonesia (1997-2015)

Thierry Fanin[1] and Guido R. van der Werf[1]

[1] Faculty of Earth and Life Sciences, Vrije Universiteit Amsterdam, Amsterdam, The Netherlands

*Correspondence to*: T. Fanin (t.fanin@vu.nl)

**Abstract**

Over the past decades, fires have burned annually in Indonesia, yet the strength of the fire season is for a large part modulated by the El Niño Southern Oscillation (ENSO). The two most recent very strong El Niño years were 2015 and 1997. Both years involved high incidences of fire in Indonesia. At present, there is no consistent satellite data stream spanning the full 19-year record, thereby complicating a comparison

between these two fire seasons. We have investigated how various fire and precipitation datasets can be merged to better compare the fire dynamics in 1997 and 2015 as well as intermediary years. We combined night-time active fire detections from the Along Track Scanning Radiometer (ATSR) World Fire Atlas (WFA) available from 1997 until 2012 and the night-time subset of the Moderate Resolution Imaging Spectroradiometer (MODIS) sensor from 2001 until now. For the overlapping period, MODIS detected

about 4 times more fires than ATSR, but this ratio varied spatially. Although the reasons behind this spatial variability remain unclear, the temporal correlation for the overlapping period was high ($R^2$=0.97) and allowed for a consistent time series. We then constructed a rainfall time series based on the Global Precipitation Climatology Project (GPCP, 1997-2015) and the Tropical Rainfall Measurement Mission Project (TRMM, 1998-2015). Relations between antecedent rainfall and fire activity were not uniform in

Indonesia. In southern Sumatra and Kalimantan, we found that 120 days of rainfall accumulation had the highest correlation with annual fire intensity. In northern Sumatra, this period was only 30 days. Thresholds of 200mm and 305mm average rainfall accumulation before each active fire were identified to generate a high fire year in southern Sumatra and southern Kalimantan, respectively. The number of active fires detected in 1997 was 2.2 times higher than in 2015. Assuming the ratio between night-time and total active

fires did not change, the 1997 season was thus about twice as fierce as the one in 2015. Although large, the difference is smaller than found in the Global Fire Emissions Database (GFED). Besides different rainfall amounts and patterns, the two-fold difference between 1997 and 2015 may be attributed to a weaker El Niño and neutral IOD conditions in the later year. The fraction of fires burning in peatlands was higher in 2015 compared to 1997 (61% and 45%, respectively). Finally, we found that the non-linearity between

rainfall and fire in Indonesia stems from longer periods without rain in extremely dry years.

Keywords: night-fires, Rainfall, Indonesia, Peatland, El Niño



## 1 Introduction

High forest clearing rates, drainage of carbon-rich peatlands, and lack of compliance with existing laws have made Indonesia the country with the highest density of fire emissions world wide. These fires impact air quality (Johnston et al., 2012; Marlier et al., 2013) and establish Indonesia as one of the world's largest greenhouse gas emitters when combining fossil fuel, deforestation, fire, and peat oxidation emissions (Andres et al., 2011; Hansen et al., 2013; van der Werf et al., 2008a; Hooijer et al., 2010). In Sumatra, large fires have occurred since at least the 1960s, while in Kalimantan (the Indonesian part of Borneo), large fires began to occur two decades later, most likely due to changes in land use and population density (Field et al., 2009). After 1996, Kalimantan's fire condition deteriorated when the Indonesian government initiated the Mega Rice Project (MRP). The objective was to convert one million hectares of tropical swamp forest to rice paddy fields and promote transmigration. This resulted in large scale deforestation and peatland drainage mostly in the province of Central Kalimantan (Putra et al., 2008).

Fires are generally not a natural occurrence in tropical rain forests due to their high humidity. This is especially true for most of Indonesia, which is located on the equator and therefore most of the country has high rainfall rates year-round and thus low natural fire incidences. Nevertheless, human-induced fires are frequently used as an inexpensive method for clearing forest and maintaining an open landscape. In 1997-98, the last very strong El Niño event recorded before 2015, $117,000km^2$ was burnt according to Tacconi (2003) and $100,000km^2$ according to GFED4s (Randerson et al., 2012; Giglio et al., 2013). Biomass burning fuel consumption reached up to $20kgC$ per $m^{-2}$ burned (Page et al., 2002).

The high fuel consumption stems from the extended peat deposits in Indonesia. Peat is a layer of partially decomposed plant material with a high proportion of organic matter. Tropical peatlands represent one of the major near-surface terrestrial organic carbon reserves. Peat thickness in Indonesia can extend to 20m (Page et al., 2002). Page et al. (2007) estimated the area of peatland in Indonesia to be between 168,250 and $270,000km^2$, storing between 10 to 32 Pg (Petagram or $10^{15}$ gram) of carbon, although other studies have produced higher values (Jaenicke et al., 2008). Peatlands on Sumatra represents 15% of the island surface area (Wahyunto et al., 2003), while for Kalimantan it is 11% (Wahyunto et al., 2004). Peatlands support the growth of peat swamp forests, which rarely burn under natural conditions. Drainage and deforestation causes peatlands to become drier and more susceptible to fires (Page et al., 2002). The depth of burning into the peat layer varies. Ballhorn et al. (2009) reported that in 2006 the average depth of burning was 33±18 cm for a study region in Central Kalimantan, and field reports found burning depth up to 1.5m in certain locations (Boehm et al., 2001). Peat fires release a large amount of carbon. During the 1997 event, within a 2.5 million hectare area in central Kalimantan, 0.19-0.23 Pg of carbon was released from peat combustion and 0.05 Pg from overlying vegetation according to Page et al. (2002). These peat fires that burn predominantly in the smoldering phase are the main reason for deteriorated air quality. A study conducted between 2002 and 2010 in Palangkaraya, located in central Kalimantan, showed that the longest hazardous air pollution episode occurred in 2002 and lasted for about 80 days (Hayasaka et al.,



2014). By 2010, 23,000km$^2$ of peatswamp forests were clear-felled, becoming degraded lands (Koh et al., 2011).

Indonesia has high inter-annual fire variability, closely related to the El Niño Southern Oscillation (ENSO) and the Indian Ocean Dipole (IOD) (Saji et al., 1999; Diaz and Markgraf, 2000; Field and Shen, 2008). ENSO represents a mode of inter-annual variability in the Earth system related to the strength of the Walker circulation governing sea surface temperatures in the tropical Pacific ocean (Rasmusson and Wallace, 1983). An El Niño event corresponds to the warm phase of ENSO, La Niña to the cool phase

(Trenberth, 1997). El Niño, coupled with reversed Walker circulation, leads to drought over most of Indonesia, while La Niña events bring increased rainfall. El Niño and La Niña can be classified in various ways, one approach relies on the magnitude of the sea surface temperature anomalies over the equatorial pacific region: Weak (with a 0.5° to 0.9° sea surface temperature anomaly), moderate (1.0° to 1.4°), strong (1.5° to 1.9°) and very strong (≥ 2.0°) (Zopf et al., 1978). IOD is another air-sea coupled climate mode

active in the tropical Indian Ocean. The Indian Ocean Dipole Mode Index (IODMI) was established to quantify IOD. It was defined using the difference in sea surface temperature between the tropical western Indian ocean and the tropical south-eastern Indian Ocean (Saji et al., 1999).

Rainfall sensitivity to ENSO and IOD varies throughout Indonesia, and to study these sensitivities, Indonesia is often separated into three climatic regions (Aldrian and Dwi Susanto, 2003). ENSO is an

important driver for droughts on Kalimantan (Harrison, 2001). There, 58% of the variability in mean July-November precipitation between 1948 and 2006 could be explained using a linear model based on Niño 3.4 (region 5°N–5°S, 120°W–170°W) (Field et al., 2009). Rainfall on Sumatra, on the other hand, was explained for 61% by IOD (Field et al., 2009). According to Aldrian and Dwi Susanto (2003) Sumatra itself is again divided into two climatic regions. Both of them have varying sensitivity to the two climate

modes (Iskandar et al., 2013).

Using NOAA AVHRR Global Area Coverage (GAC) Earth Observation satellite data, Wooster et al. (2012) found two to three times more active fires in Kalimantan during the 1997-98 very strong El Niño event than in the next largest fire years (e.g., 2002) due to extreme drought then. Fires do not solely occur during strong ENSO events. In 2006 for example, with only a weak El Niño, MODIS active fire (Giglio,

2010) data detected about 149,000 fires over the whole of Indonesia, against 151,000 in the 2015 very strong El Niño year. Most of Indonesian fires burn on Kalimantan and Sumatra. In 2006, both regions generated 84% of all fires in Indonesia (47% and 37%, respectively, Yulianti et al. (2012)). Although Kalimantan has higher emissions, Sumatra contributes more to smoke concentrations in populated areas in Southeast Asia outside of Indonesia due to its location west of the Malaysian peninsula (Marlier et al.,

2015) making Singapore and Malaysia most vulnerable to fire emissions from Sumatra (Hyer and Chew, 2010; Salinas et al., 2013; Aouizerats et al., 2015). Under future climate change scenarios and without changing fire management practice, Indonesia could become even more vulnerable to fire (Herawati and Santoso, 2011).



Previous studies have investigated the relationship between precipitation and fires. Field and Shen (2008), using cross-calibrated ATSR, TRMM, and MODIS active fire detections and the Global Precipitation Climatology Project (GPCP) data set for rainfall for the 1997-2006 period, showed that there was a risk of severe burning when, for a period of 4 months, precipitation falls below a threshold of 350mm in southern Sumatra and 650mm in southern Kalimantan. Van der Werf et al. (2008a) found a strong nonlinear relation between fire and rainfall, where active fire detections increased exponentially below 100 mm month$^{-1}$ of rainfall during the dry season. Using visibility records from Sumatra and Kalimantan's World Meteorological Organization level meteorological stations, Field et al. (2009) calculated monthly mean extinction coefficients between 1960 and 2006. The optimal timeframe to explain variances in extinction coefficients on Sumatra was after 5 months of rainfall accumulation. With a piecewise linear regression, this could explain 67% of the observed patterns with a threshold of 609mm, between 1960 and 1983, and 85% with a threshold of 631mm between 1984 and 2006 on Sumatra. On Kalimantan, 4 months' rainfall accumulation explained 78% of the extinction coefficient variances with a threshold of 672mm between 1984 and 2006.

For the last two decades, the large fire event of 1997-98 has frequently been used as a reference point. Since then, fire years have been relatively low, except for moderate or weak El Niño events (2002, 2006, 2009, 2012). The amount of fires in 2013 was below our study period average, but northern Sumatra experienced a large amount of fires over a very short period of time (Gaveau et al., 2014). In 2014, fire activity was for the first time above average despite that year being ENSO neutral followed by the very strong El Niño event of 2015 with disruption of regular life in large parts of Indonesia due to dense smoke from the large number of fires. The aim of this study is to better understand the fire-precipitation dynamics in general and focus on the dynamics of 2015, 2014, 2013, and 1997 in particular, based on a merged but consistent fire and rainfall dataset.

## 2 Datasets and Methods:

Our study region includes all of Indonesia, but we focused primarily on the islands of Sumatra and Kalimantan, as they generate most of the fire activity within the country (fig 1). Sumatra is situated between 6°N-6°S and 95°E-107°E. Most of the island has been deforested, but primary forests remain on the mountainous coastal range of the island facing the Indian Ocean. The other side of the island with the provinces of Riau, Jambi, and South Sumatra is largely covered with degraded forests and plantations mostly for palm oil and for pulp and paper production. Peatlands are abundant in this region, especially in the low-lying areas towards the Northeast. Kalimantan is the Indonesian part of Borneo. It is located between 5°N-4°S and 108°E-118°E. Intact forest remains in the northern part of Kalimantan, but most of southern Kalimantan consists of degraded forest or plantations. A large scar of deforested land is located in the south, due to the Mega Rice Project (Aldhous, 2004). This widespread area contains some of the deepest peat in Indonesia but has been extensively drained.



To better understand the rainfall patterns preceding and coinciding with the fire season, we based our
analysis on a somewhat modified division of the three climatic regions identified by Aldrian and Dwi
Susanto (2003) to better account for differences in fire patterns: Their first region is the southern part of
Indonesia, from southern Sumatra to Timor island, southern Kalimantan, Sulawesi and part of Irian Jaya.
Their second is northwest Sumatra and northwest Kalimantan, and finally the last one includes Maluku and
northern Sulawesi. We separated their first region into southern Sumatra (SoSu) and southern Kalimantan
(SoKa). Our third region is Sumatra north of the equator (NoSu). Finally, their last region and the area not
included in our analysis was classified as "Other" due to less fire activity (fig 1).

One key issue when aiming to compare 1997 with 2015 is that there is no fire dataset covering the full
period of time based on one satellite stream. There is one for precipitation, but its resolution is relatively
coarse (1° spatial resolution). To obtain rainfall data across Indonesia for the 1997-2015 period we
combined two datasets: for 1997, the One-Degree Daily (1DD) product from the Global Precipitation
Climatology Project (GPCP, (Huffman et al., 2001)) and from 1998 until 2015, the Tropical Rainfall
Measurement Mission Project (TRMM) 3B42 data version 7 (Simpson et al., 1996; Huffman et al., 2007).
We merged these two datasets (explained below) to have the higher spatial resolution from TRMM for
most of the time period and still cover the full time period. TRMM is based on a combination of TRMM
precipitation radar (PR) and TRMM microwave imager (TMI) and rain gauge measurements at a 0.25°
spatial resolution (Kummerow et al., 1998; Islam and Uyeda, 2007). It lost radar data after mid-2015.
TRMM 3B42 has a 3-hourly temporal resolution available since 1998. For the purpose of this study,
TRMM 3B42 data was cumulated to daily temporal resolution.

Daily GPCP uses the Threshold Matched Precipitation Index (TMPI) and geostationary infrared satellite
data to determine the daily rainfall rate at 1° spatial resolution from the monthly data produced from gauge
stations, satellites and sounding observations (Pendergrass, 2015). To aggregate 1997 GPCP with the
remaining TRMM, we calculated a correction factor for each 1° grid cell derived from a linear regression
between GPCP and TRMM based on the 4 driest months of each year between 1998 and 2014. We then
used the slope to make the 1997 GPCP rainfall data consistent with the TRMM data. We found a slope
relatively homogenous over the country (average slope over Indonesia: 1.1) (fig 2). The correlation showed
large spatial variability. South of the equator, correlations were high; with most areas having $R^2$ values
exceeding 0.8. North of the equator and in mountainous regions, $R^2$ was lower, for a large part related to
lower interannual variability here, but still exceeded 0.5 in most areas.

To cover our study period we also had to merge two fire datasets. We used the Along Track Scanning
Radiometer (ATSR) World Fire Atlas (WFA) algorithm 2 (Arino et al., 2012) available from 1995 until
2012, and the Thermal Anomalies/Fires product MOD14A1 (Terra) from the Moderate Resolution Imaging
Spectroradiometer (MODIS) available from 2001 until 2015 (Justice et al., 2002). The ATSR fire algorithm
(in the remainder referred to as ATSR) detects active fire data at 10:30 PM local time, avoiding flagging
hot surfaces or solar reflection as fire. Using the 3.7-micrometer band, it observes hot spots warmer than



308 Kelvin. Because of night-time detection, it underestimates the total amount of fires due to the strong diurnal cycle of fires with in general a peak in the afternoon (Giglio, 2007).

Terra acquires data twice daily (10:30 AM and PM) at 1km spatial resolution using three different tests to identify fire activity. The first is an absolute threshold test (Kaufman et al., 1998), where grid cells with temperatures higher than 360K during daytime or 320K at night are potentially fires. The second is a
background characterisation, the neighbouring pixels of the potential fire pixel are used to estimate a background value. By using different statistical techniques; it uses neighbouring pixels to identify fire activity comparing the potential fire pixel with the background value to flag the pixel as burned or to eliminate as false alarms. Finally, if previous tests are successful, it looks for characteristics in the active fire signature with the 4μm brightness temperature (T4) to find variations from non-fire backgrounds
applying various thresholds (Giglio et al., 2003). These methods are highly effective at identifying smaller fires, yielding near-time information (Roy et al., 2008).

For consistency in data acquisition time, we limited our use of Terra data to night-time (10:30PM) to have late evening fires at nominal and high confidence. While MODIS revisits the same site every day, ATSR overpasses the same grid cell only once every 3 days (Mota et al., 2006). To combine ATSR and Terra we
used a correction factor based on the fire detection ratio between monthly ATSR and Terra observations for the period of 2001 and 2011 averaged over 1° grid cells. We then multiplied the ATSR data with the correction factor for the period 1997 – 2000 for all grid cells within the 1° grid cells. This correction factor showed substantial spatial variability. In most high fire regions, ATSR was around 4 times lower than Terra, although in some grid cells the difference was a factor 12 (fig 3). Our fire correction factor
compensated for the lower sampling rates from ATSR over Terra between 2001 and 2011 in all Indonesia ($R^2$ is 0.97). We have not used the burned area or emissions dataset from the Global Fire Emissions Database (GFED) because it relies on the ratio between all MODIS data and ATSR, not just the night time subset.

Previous work has shown that 4 months of rainfall accumulation has the best correlation to identify the
potential risk of high fire activity in southern Sumatra and Kalimantan (Field and Shen, 2008). However, this period may be different in other regions including northern Sumatra because of different rainfall patterns (Aldrian and Dwi Susanto, 2003). We therefore investigated different lengths of rainfall accumulation to identify the most appropriate time window for different regions. Using the daily rainfall data, we calculated the build-up of rain prior to each fire on time frames of 0, 7, 15, 30, 60, 90 and 120
days. In our analyses we used two additional datasets. To better understand the diurnal fire dynamics, we calculated the fraction of daytime fires burning overnight between 2003 and 2015. Daytime fires were acquired from the Thermal Anomalies/Fires product MYD14A1 (Aqua) from MODIS. It uses the same detection algorithm as Terra mentioned above but with an acquisition time at 2.30 PM and AM. To estimate the number of daytime active fire detections, we only kept the 2.30 PM fires. Finally, we used the



Wetlands International (WI) PeatAtlas of Sumatra, Kalimantan and Papua (Wetlands International, 2015)
to better understand peat fire dynamics.

## 3 Results

Indonesia's position on the equator, surrounded by two ocean basins, caused substantial regional variability
in the fire season. In southern Sumatra and Kalimantan, the dry season lasted from June until heavier
rainfall started again sometime in October for a total of 4-5 months (fig 4). During our study period, the
months with the highest incidence of active fires were September and October. Average rainfall during the
dry season was 140mm per month in southern Sumatra, compared to 170mm in southern Kalimantan, but
with substantial regional variation. Northern Sumatra, on the other hand, had two fire seasons, a first one in
February – March (early dry season) followed by a second in June lasting up to 3 months, depending on the
year (late dry season).

The dataset covering the 1997- 2015 period enabled us to observe several large fire years, including two
during very strong El Niño events (1997 and 2015). In 1997, approximately 119,000 active night-time fire
detections were made according to our merged dataset, compared to less than half that number in 2015. The
number for 1997 was to some degree a virtual number based on the observed ATSR fires and the correction
factor to bring the ATSR fires in line with those observed by Terra at night. In the remainder of this paper,
we will refer simply to the number of active fire detections instead of the number of night-time fires from
the merged product. These two extreme drought years mainly affected southern Sumatra and southern
Kalimantan, with approximately 50,000 active fires in 1997 and 20,000 active fires in 2015 in both regions
(fig 5). Other relatively high fire years in these two regions occurred during weak or moderate El Niño
events in 2002, 2006, 2009 and 2014.

Northern Sumatra's inter-annual variability was very different from that of the other regions. Years of
strong active fire activity did not always line up with El Niño events as shown earlier by Gaveau et al.
(2014). Fires there were also less frequent than in southern Sumatra and Kalimantan. Nevertheless, in 2005
and 2014 more than 10,000 active fires were detected in northern Sumatra. Northern Sumatra had a large
proportion of active fire detections in peatlands (table 1). During our study period, on average 75% of the
total active fires occurring in northern Sumatra were on peatland, with higher values of 87% and 88% in
2005 and 2014, respectively. In Indonesia south of the equator, the proportion of peat active fires was
lower, averaging 61% in southern Sumatra and 50% in southern Kalimantan. Between 1997 and 2015, our
dataset indicated 192,000 active fires in the 127,425km$^2$ of peatland in Sumatra and Kalimantan.

The number of active fires burning through the night across Indonesia (expressed as the ratio between
night-time and daytime active fires) varied from 11% in 2003 up to 36% in 2015, on average 20%.
Northern Sumatra, although having fewer active fires than the southern regions, had between 14% (low fire
year) up to 52% (high fire year) of daytime fires burning overnight, and on average 33%. Southern Sumatra
and southern Kalimantan had a lower proportion of fires burning through the night with years as low as 4%,



but on average 23% and 22% respectively. In both of these regions, 2015 had the highest proportion of fires burning through the night with 53% and 43% respectively (table 2).

Using daily rainfall and active fires, we observed that in southern Sumatra and southern Kalimantan, the fire season and dry season (defined here as the period with the 120 driest days) followed relatively similar patterns on both islands but still had important differences. In 1997, the dry season started in early July on

both islands, but while days with more than 500 active fires began in mid-August in southern Kalimantan, the fire season in southern Sumatra began almost one month later, potentially because rainfall still occurred in Sumatra (fig 6). On both islands most active fires burned during the driest 120 days. 2006 had a weak El Niño, with more days with rainfall during the dry season than in 1997. In southern Sumatra, strong rainfall in late October ended the fire season earlier in that year than in 1997. In southern Kalimantan, the late

October rain did not exceed 8 mm/day, which enabled further burning before the return of the monsoon. In 2015, the dry season as defined here began earlier than in 1997 while active fires started mildly in mid-August, rising in September for about two months, as had occurred in 1997. During the dry season, however, there were more days with rainfall and the number of fires detected was lower than in 1997. Furthermore, in 2015 active fires stopped already late October, while in 1997 they burned for another week

in November.

The amount of rainfall during the dry season did not influence the number of active fires equally in the different regions. In other words, there was large spatial variability in the vulnerability to fire. In 1997, all of Indonesia was dry, but most active fires occurred south of the equator, northern Sumatra being only mildly affected by fires. Obviously, all high fire years in southern Sumatra and southern Kalimantan had a

relatively low amount of rainfall during the entire dry season, but rainfall amount differed between the high fire years. While in 1997 no grid cell in these two regions exceeded 15mm of rain per month during the driest 120 days period, 2015 had highs at 38mm. Northern Sumatra had rainfall accumulation of 194mm during the dry season in 2005, while 2014 was drier with only 147mm (fig 7).

Observations of the accumulation of rain before active fires showed that in all regions studied extreme fire

years had less than 1mm of rain on the day of fire detection (fig 8). Yet, this also applies to some very mild fire years and is therefore not a good predictor for high fire years. In line with previous studies, we have therefore examined up to 120 days of rainfall preceding active fire observations in order to observe the influence of rainfall on fires. The novelty in our approach is that we used daily data for this instead of monthly. In southern Sumatra and southern Kalimantan this did not lead to new results; cumulative rainfall

for 120 days prior to active fires were the most effective to identify fire vulnerability to rainfall. On the other hand, in northern Sumatra, a 30-day timeframe was more appropriate (table 3). This shows that northern Sumatra is more responsive to relatively short-term drought periods, or that droughts last shorter.

This method allowed us to observe rainfall thresholds to generate high fire years. In northern Sumatra, we found that years with less than 120mm of cumulated rainfall within 30 days prior to active fires could

generate a high fire year. Analysing high fire years in 2005 and 2014, we found substantial differences. In



2005, 93mm of cumulated rainfall prior to active fire detections was measured against only 33mm in 2014. This shows that the vulnerability of fires to rainfall is harder to evaluate in this region. For example, in 1997, with 12mm less cumulated rainfall within 30 days prior to active fires than in 2005, northern Sumatra had 42% less burning, highlighting the role of other factors. In Sumatra south of the equator, both 1997 and 290 2015 were high fire years with less than 200mm of cumulated rainfall within 120 days prior to active fires. For these two years in Kalimantan, the threshold was higher at around 300mm. In both southern regions we observed that 1997 burned much more than 2015, with in absolute terms a relatively small difference in rainfall accumulation prior to active fires (118,685 and 53,659 active fires respectively).

**4 Discussion**

The use of a correction factor to merge two rainfall and fire datasets enabled us to obtain a relatively consistent 19-year time series covering two very strong El Niño episodes. Interannual variability in fire activity in Indonesia was not solely linked to ENSO and varied between regions. The regions that were most tightly linked to droughts associated with El Niño events, such as in 1997 and 2015, were those south of the equator in Sumatra and Kalimantan. We found that 2015 had only about half the number of active 300 fires observed in 1997 (fig 5). This large difference came mostly from lower rainfall in 1997 between mid-July and mid-September. The dry season in the areas south of the equator lasted for approximately 4 months in those two years but 2015 saw more relatively minor rain events of a few mm day$^{-1}$ during the dry season (fig 6). These minor events could be one of the causes of the strong non-linearity found in previous studies (van der Werf et al., 2008b); they add relatively little rainfall but are apparently capable of pausing 305 the fire season for some time.

We have shown that in southern Sumatra and southern Kalimantan rainfall accumulation over 120 days had the best correlation to estimate the number of fires observed. The amount of rainfall during the transition between the wet and dry season was essential to estimate the beginning of the fire season. The El Niño during this transition period was stronger in 1997 than in 2015 and may be part of the explanation for 2015 310 having only half the number of active fires detected in 1997 (fig 9). Although both years were very strong El Niño years, their temporal dynamics were different. In 2015, the highest sea surface temperature anomaly in the Pacific occurred in September (2.5°C). In 1997, however, anomalies were already at 2.8°C in July, rising to above 3°C in August and September. A second difference was that the Indian Ocean temperature anomalies in 1997 were twice as high as in 2015. Here we can only show that the ENSO and 315 IOD dynamics were different between the two years and hypothesize this to be the root cause for the difference in rainfall. Another factor in explaining the large difference in number of active fires detected may be that the 1997 event occurred just after the Mega Rice Project, which increased the flammable area substantially. However, according to Field et al. (2015) Kalimantan as a whole has become even more vulnerable to fire over the years.



While in the southern hemisphere IOD and ENSO dynamics and their impact on 120-day rainfall amounts could explain the variability in active fires detected between the various years, in northern Sumatra, the number of active fires annually detected was more closely correlated with shorter periods of rainfall. The dry season in general lasts shorter there, but northern Sumatra has two of them in some years (Yulianti et al., 2013) making comparisons between various years difficult. 2005 and 2014 saw active fires during both

dry seasons but the early dry season had most fires (fig 6). In 2013, burning only occurred in the late dry season with the months of May and June being drier than usual. This short period contained most of the yearly fire activity and generated more atmospheric pollution over Singapore than the 1997 fires because the winds carry smoke from this region directly to Singapore (Gaveau et al., 2014).

We observed that in northern Sumatra more active fires burned overnight than in southern Sumatra and

Kalimantan. The southern regions thus had a relatively low fraction of active fires burning overnight, except in 2015, coinciding with a very strong El Niño event. Unfortunately, due to data availability we do not know whether the same holds for the 1997 El Niño (table 2).

Having focused exclusively on active fires detected at night, this study is not readily comparable with Field and Shen's (2008) rainfall accumulation threshold of 350mm in southern Sumatra and 650mm in southern

Kalimantan. Conditions at night are less favourable for fires, the fires that do burn overnight are mostly intense fires and are probably better correlated with rainfall accumulation vulnerability. In southern Sumatra, the threshold of 350mm within 4 months prior to active fire detected by Field and Shen's (2008) was valid for high fire years. While our estimate of 200mm seems much lower, there were no years with rainfall accumulation in between 200mm and 350mm rainfall accumulation within 4 months prior to active

fire (fig 8).

This study relied exclusively on night-time data in order to remain consistent with the datasets used to cover the entire study period. One uncertainty for this methodology is an eventual change in diurnal cycle within similar climatic and ENSO conditions. We found that 36% of active day fires burned overnight in 2015, compared to only 21% in 2006, yet 2006 had a much weaker El Niño. Unfortunately, no consistent

day-time fire data was available to evaluate the diurnal cycle extending back to 1997.

Our approach carries a number of uncertainties. First, the focus on night time fires enabled a consistent time series but, as mentioned above, potential changes in the number of fires observed during the day without impacting the number of night time fires (i.e. a change in the number of fires that burn through the night) would remain undetected. Whether this is the reason behind the different ratio between 1997 and

2015 found here (just over 2) and in the GFED carbon emission estimate (almost 3) cannot be resolved. Second, active fire datasets have a large omissions rate. Tansey et al. (2008) compared MODIS Terra data (MOD14A1) hotspots to with disaster monitoring constellation (DMC) and Landsat TM data. They found an omission error of 60% during the 2002, 2004 and 2005 dry seasons. The advantage of using active fires is the high temporal resolution, which enabled us to better study fire-precipitation rates. Finally, our

approach of stitching together different datasets revealed the substantial spatial variability in ratios mostly



between ATSR and MODIS active fires for which we do not have a plausible explanation but potentially impacted the time series. In addition, the lower sampling rate of ATSR means that the daily time step for 1997 is actually a constructed one, the actual data has a 3 days revisit time.

## 5 Conclusions:

We have constructed a rainfall and active fire dataset by combining two datasets for each to cover the 1997 – 2015 period thus capturing two strong El Niño events. The dataset provides daily temporal resolution and focused on night-time fires to provide consistency over the full time period. We analysed the fire patterns in Indonesia by identifying 4 regions with different spatio-temporal variability in fire activity. The main findings can be summarized as follow:

-     While most fires in southern Sumatra and Kalimantan occurred between August and October, northern Sumatra had two fire seasons: a short one in February followed by a longer one between June and August. In high fire years, most active fires occurred in the early dry season, except in 2013 when almost all active fires occurred during the later dry season.

-     The 2015 El Niño event involved approximately 52,000 active fires, compared to approximately 119,000 in 1997. This difference came mostly from southern Sumatra and Kalimantan. While both 1997 and 2015 were very strong El Niño's, the Indian Ocean Dipole (IOD) was more favourable for droughts and the El Niño event in 1997 occurred somewhat earlier, thus aligning better with the start of the fire season.

-     While amounts of rainfall in 1997 and 2015 were relatively similar 120 days prior active fires (respectively 281mm and 303mm in southern Sumatra and 248mm and 269mm in southern Kalimantan), the number of days with low rainfall in 2015 was much larger than in 1997. This is one of the key reasons behind the strong non-linear response of fires to rainfall; while those days with little rainfall do not add much to the total dry season rainfall, they are capable of halting fires.

-     Northern Sumatra, with approximately 42,000km$^2$ of peatland, had the highest proportion of active fires observed in peatland (75%), compared to 61% in southern Sumatra. Although southern Kalimantan has the largest peatland area, with more than 50,000km$^2$, the incidence of active peat fires in that region was 50% of all active fires, which is also the average percentage when considering all of Indonesia.

-     -In agreement with other studies, we found that 120 days of rainfall accumulation prior to an active fire was the best correlated timeframe to a severe fire year in southern Sumatra and Kalimantan. In northern Sumatra this period was lower, 30 days. Part of the reason is that the dry season there is shorter. However, the response time is faster than in other parts of Indonesia as well.

## Acknowledgements:

This research was supported by the European Research Council, grant number 280061.





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



**Tables**

Table 1: Number of active fire detections for the different regions including the number of fires detected in peatlands , with percentage of peatland. For each region the total surface of peatland is shown on top of each column.


| Year | NoSu (41,792 km$^2$) | | | SoSu (35,546 km$^2$) | | | SoKa (50,087 km$^2$) | | | Others (100,441 km$^2$) | | | Total (227,866 km$^2$) | | |
|---|---|---|---|---|---|---|---|---|---|---|---|---|---|---|---|
| | Peat fire | Non-peat fire | % | Peat fire | Non-peat fire | % | Peat fire | Non-peat fire | % | Peat fire | Non-peat fire | % | Peat fire | Non-peat fire | % |
| 1997 | 4,031 | 2,881 | 58 | 24,645 | 21,842 | 53 | 22,624 | 30,349 | 43 | 2,228 | 10,085 | 18 | 53,528 | 65,157 | 45 |
| 1998 | 1,714 | 1,322 | 56 | 138 | 378 | 27 | 599 | 3,126 | 16 | 873 | 19,952 | 4 | 3,324 | 24,778 | 12 |
| 1999 | 1,691 | 990 | 63 | 918 | 841 | 52 | 494 | 654 | 43 | 103 | 1,111 | 8 | 3,206 | 3,596 | 47 |
| 2000 | 1,624 | 2,496 | 39 | 215 | 564 | 28 | 146 | 332 | 31 | 83 | 904 | 8 | 2,068 | 4,296 | 32 |
| 2001 | 1,287 | 553 | 70 | 257 | 243 | 51 | 1,600 | 766 | 68 | 54 | 1,187 | 4 | 3,198 | 2,749 | 54 |
| 2002 | 3,720 | 1,063 | 78 | 1,220 | 697 | 64 | 7,336 | 5,836 | 56 | 134 | 3,579 | 4 | 12,410 | 11,175 | 53 |
| 2003 | 1,582 | 685 | 70 | 1,054 | 557 | 65 | 1,125 | 1,249 | 47 | 17 | 996 | 2 | 3,778 | 3,487 | 52 |
| 2004 | 2,138 | 915 | 70 | 2,100 | 786 | 73 | 2,543 | 3,497 | 42 | 200 | 2,389 | 8 | 6,981 | 7,587 | 48 |
| 2005 | 9,627 | 1,497 | 87 | 559 | 187 | 75 | 1,032 | 770 | 57 | 226 | 1,402 | 14 | 11,444 | 3,856 | 75 |
| 2006 | 2,649 | 741 | 78 | 5,518 | 2,872 | 66 | 8,309 | 8,521 | 49 | 394 | 2,418 | 14 | 16,870 | 14,552 | 54 |
| 2007 | 968 | 146 | 87 | 219 | 519 | 30 | 90 | 409 | 18 | 91 | 1,175 | 7 | 1,368 | 2,249 | 38 |
| 2008 | 777 | 263 | 75 | 551 | 980 | 36 | 42 | 195 | 18 | 57 | 996 | 5 | 1,427 | 2,434 | 37 |
| 2009 | 2,686 | 985 | 73 | 1,831 | 1,436 | 56 | 4,928 | 4,232 | 54 | 476 | 2,529 | 16 | 9,921 | 9,182 | 52 |
| 2010 | 655 | 176 | 79 | 69 | 94 | 42 | 7 | 93 | 7 | 29 | 789 | 4 | 760 | 1,152 | 40 |
| 2011 | 1,339 | 373 | 78 | 2,191 | 1,284 | 63 | 1,116 | 914 | 55 | 280 | 865 | 24 | 4,926 | 3,436 | 59 |
| 2012 | 1,898 | 540 | 78 | 2,195 | 1,867 | 54 | 1,320 | 701 | 65 | 147 | 1,355 | 10 | 5,560 | 4,463 | 55 |
| 2013 | 5,320 | 1,133 | 82 | 263 | 791 | 25 | 357 | 196 | 65 | 106 | 957 | 10 | 6,046 | 3,077 | 66 |
| 2014 | 10,807 | 1,453 | 88 | 4,147 | 1,185 | 78 | 3,549 | 2,262 | 61 | 783 | 3,651 | 18 | 19,286 | 8,551 | 69 |
| 2015 | 1,551 | 608 | 72 | 17,312 | 5,577 | 76 | 13,227 | 7,757 | 63 | 579 | 7,048 | 8 | 32,669 | 20,990 | 61 |
| total | 56,064 | 18,820 | 75 | 65,402 | 42,700 | 61 | 70,444 | 71,859 | 50 | 6,860 | 63,388 | 10 | 395,537 | 196,962 | 67 |



Table 2: MODIS Terra (10:30pm) and MODIS Aqua (2:30pm) active fire detections.

| Year | NoSu | | | SoSu | | | SoKa | | | Others | | | Total | | |
|------|----------|------------|----|----------|------------|----|----------|------------|----|----------|------------|----|----------|------------|----|
| | day fire | night fire | % | day fire | night fire | % | day fire | night fire | % | day fire | night fire | % | day fire | night fire | % |
| 2003 | 9,442 | 2,267 | 24 | 14,430 | 1,611 | 11 | 19,895 | 2,374 | 12 | 23,406 | 1,013 | 4 | 67,173 | 7,265 | 11 |
| 2004 | 13,395 | 3,053 | 23 | 21,053 | 2,886 | 14 | 36,234 | 6,040 | 17 | 41,009 | 2,589 | 6 | 111,691 | 14,568 | 13 |
| 2005 | 28,654 | 11,124 | 39 | 8,656 | 746 | 9 | 13,649 | 1,802 | 13 | 19,983 | 1,628 | 8 | 70,942 | 15,300 | 22 |
| 2006 | 13,045 | 3,390 | 26 | 42,031 | 8,390 | 20 | 55,614 | 16,830 | 30 | 38,696 | 2,812 | 7 | 149,386 | 31,422 | 21 |
| 2007 | 5,771 | 1,114 | 19 | 13,468 | 738 | 5 | 9,133 | 499 | 5 | 16,339 | 1,266 | 8 | 44,711 | 3,617 | 8 |
| 2008 | 7,039 | 1,040 | 15 | 13,391 | 1,531 | 11 | 6,284 | 237 | 4 | 14,933 | 1,053 | 7 | 41,647 | 3,861 | 9 |
| 2009 | 12,100 | 3,671 | 30 | 19,830 | 3,267 | 16 | 36,817 | 9,160 | 25 | 29,502 | 3,005 | 10 | 98,249 | 19,103 | 19 |
| 2010 | 5,846 | 831 | 14 | 3,942 | 163 | 4 | 2,118 | 100 | 5 | 8,465 | 818 | 10 | 20,371 | 1,912 | 9 |
| 2011 | 8,425 | 1,712 | 20 | 17,498 | 3,475 | 20 | 17,249 | 2,030 | 12 | 17,992 | 1,145 | 6 | 61,164 | 8,362 | 14 |
| 2012 | 8,661 | 2,438 | 28 | 21,300 | 4,062 | 19 | 17,691 | 2,021 | 11 | 21,175 | 1,502 | 7 | 68,827 | 10,023 | 15 |
| 2013 | 14,761 | 6,453 | 44 | 9,617 | 1,054 | 11 | 12,721 | 553 | 4 | 16,440 | 1,063 | 6 | 53,539 | 9,123 | 17 |
| 2014 | 23,736 | 12,260 | 52 | 18,521 | 5,332 | 29 | 29,484 | 5,811 | 20 | 35,473 | 4,434 | 12 | 107,214 | 27,837 | 26 |
| 2015 | 6,155 | 2,159 | 35 | 43,211 | 22,889 | 53 | 49,237 | 20,984 | 43 | 52,429 | 7,627 | 15 | 151,032 | 53,659 | 36 |
| total | 157,030 | 51,512 | 33 | 246,948 | 56,144 | 23 | 306,126 | 68,441 | 22 | 335,842 | 29,955 | 9 | 1,045,946 | 206,052 | 20 |


Table 3: standard error of the regression for each region according to the number of days of rainfall accumulation.

| | 0 | 7 | 15 | 30 | 60 | 90 | 120 |
|------|--------|--------|--------|-------|-------|-------|-------|
| NoSu | 2,418 | 1,940 | 1,961 | 1,897 | 1,981 | 2,468 | 2,874 |
| SoSu | 6,638 | 8,254 | 8,579 | 7,299 | 5,933 | 5,053 | 4,550 |
| SoKa | 10,022 | 11,183 | 10,069 | 7,843 | 7,940 | 6,866 | 4,894 |



**Figures**

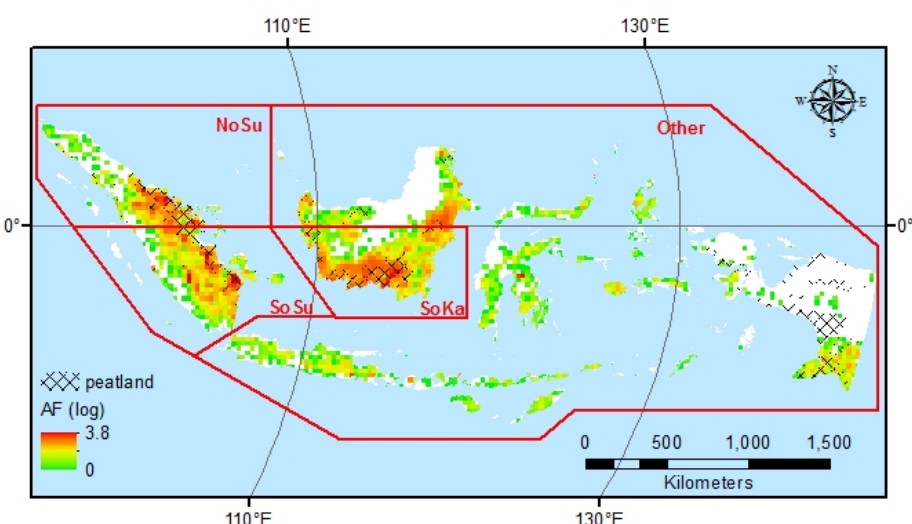

Figure 1: Region definition, active fires (log scale, rescaled to quarter degree) and peatland extent for Indonesia. NoSu is Northern Sumatra, SoSu Southern Sumatra, and SoKa is southern Kalimantan.


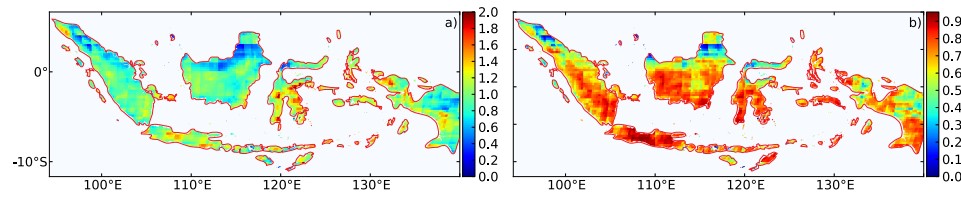

Figure 2: a) Slope and b) R² of a linear regression between GPCP and TRMM rainfall based on the cumulated rainfall for the 4 driest months between 1998 and 2014.

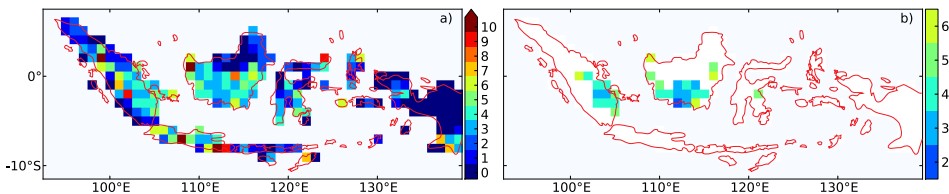


Figure 3: Ratio between MODIS Terra and ATSR (both 10:30pm overpass) active fire detections over the 2001-2011 overlapping period for a) all grid cells with active fire detections and b) only for those grid cells with more than 500 Terra active fires detected between 2001 and 2011.





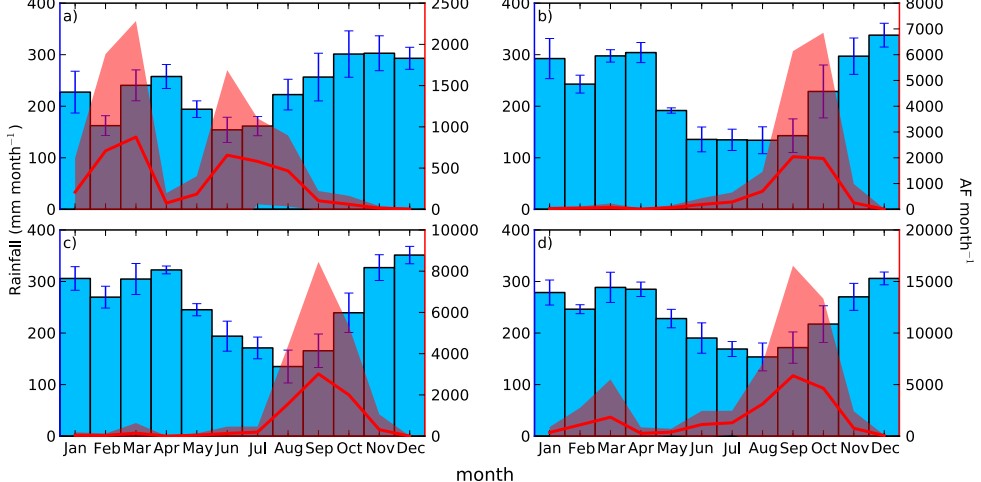


Figure 4: Monthly rainfall with standard deviation (blue) and active fire detections with standard deviation (red), averaged over 1997 -2015 for a) northern Sumatra, b) southern Sumatra, c) southern Kalimantan and d) all of Indonesia.


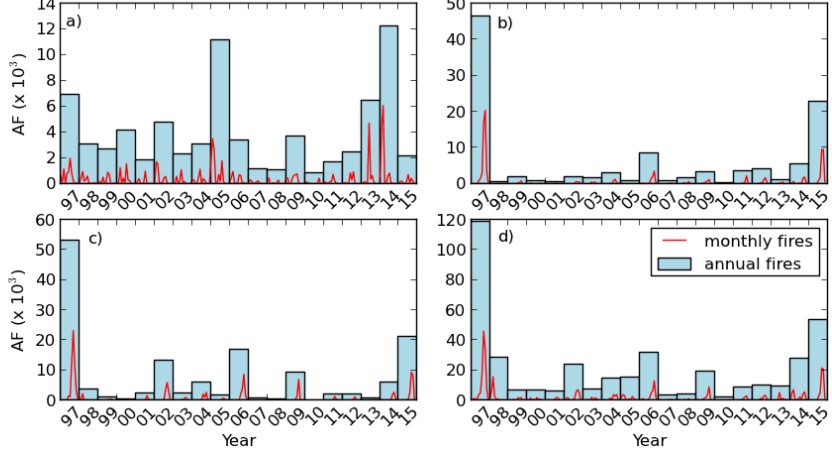

Figure 5: Annual and monthly active fire (AF) detections for a) northern Sumatra, b) southern Sumatra, c) southern Kalimantan and d) all of Indonesia






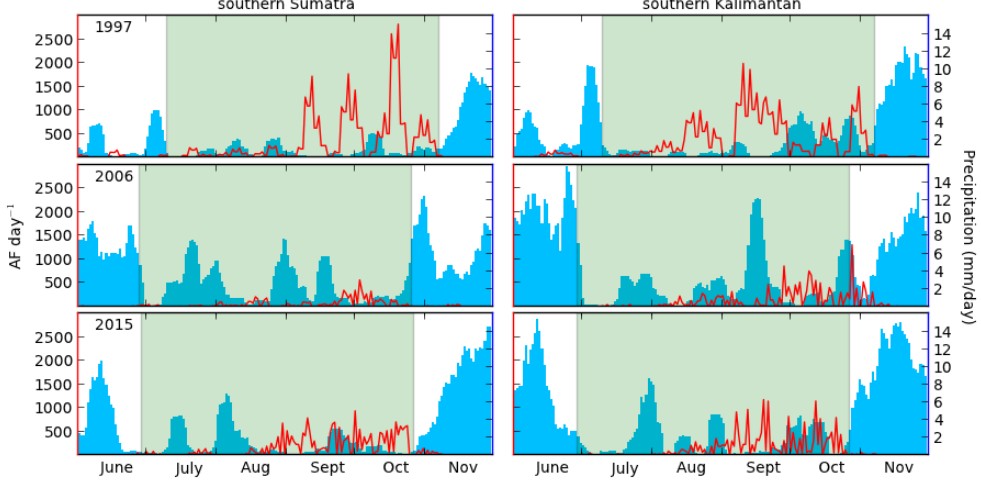

Figure 6: Daily active fire (AF) detections (red) and rainfall (blue) in southern
Sumatra and southern Kalimantan during dry seasons of 1997, 2006 and 2015. The
green shaded period indicates the dry season defined here as the 120 day period
with lowest rainfall.





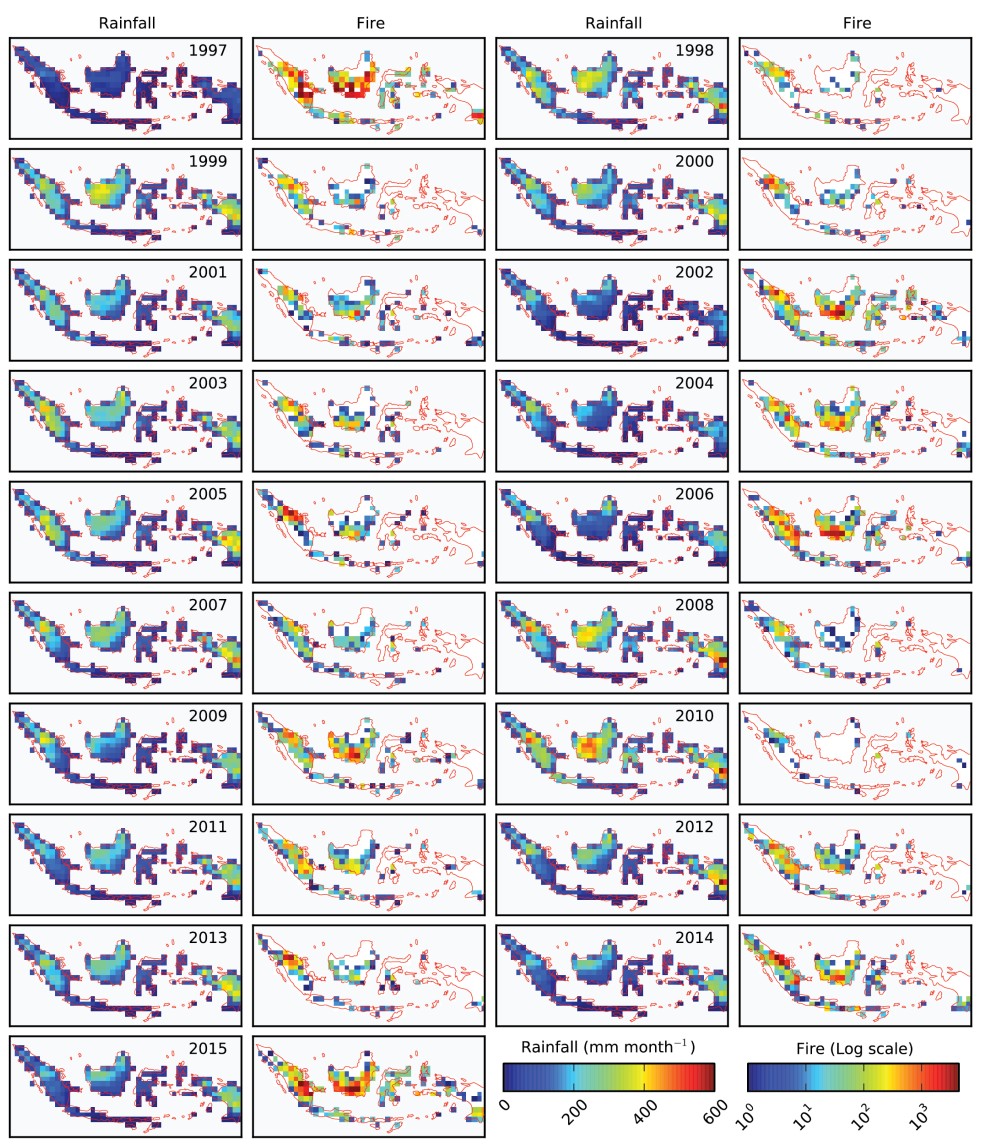

Figure 7: Average monthly rainfall and active fire detections (log scale) aggregated to 1° spatial resolution during the dry season.




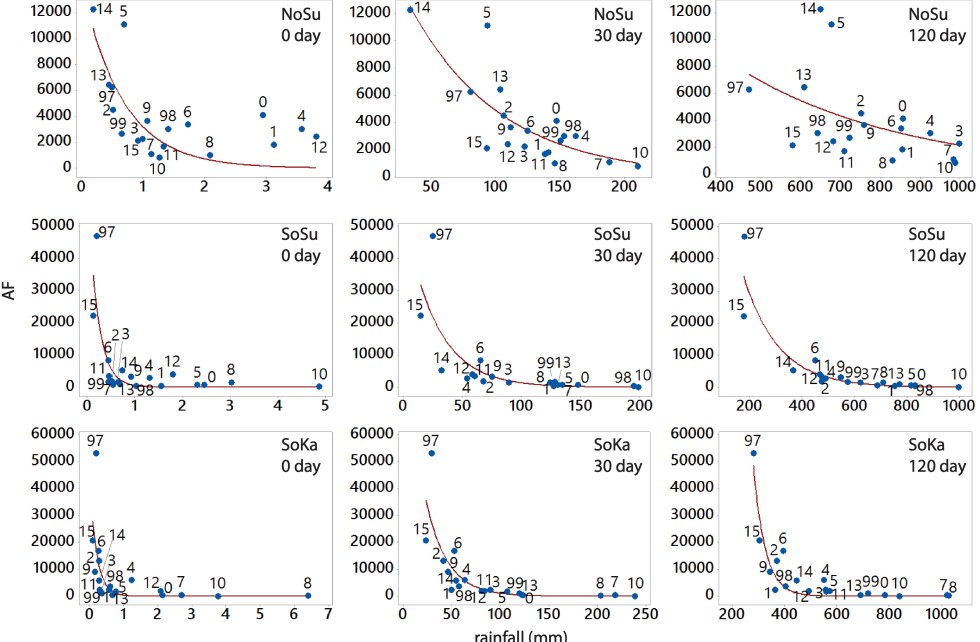

Figure 8: mean rainfall accumulation before fire (with exponential trend line) for each climatic region with a 0, 30 and 120 days buildup. The numbers in the graph denote the year (97 is 1997, 0 is 2000, through 15 is 2015).


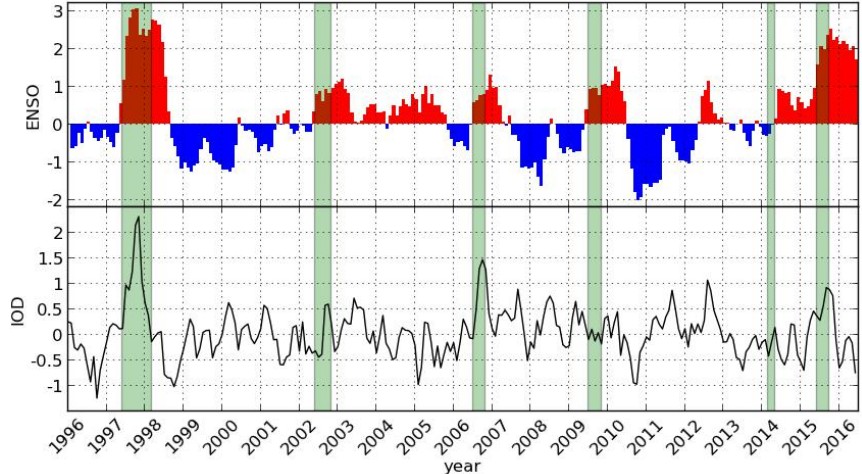

Figure 9: ENSO and IOD with periods of strongest fire activity in green.