# Peer review of "Precipitation-fire linkages in Indonesia (1997-2015)"

_Biogeosciences, 2016_

## Referee Comment (RC1) · Anonymous Referee #1 · 7 Dec 2016

General comments

The manuscript presents valuable information on the influence of precipitation on vegetation and peat fires in Indonesia, especially as two datasets each of precipitation and satellite-derived active fire detections are merged to cover the period 1997 to 2015 and thus two very strong El Niño years. While it is generally very well written, the manuscript combines a rather large number of single aspects (combining the datasets to build time series, comparing 1997 to 2015, discussing the full time series, discussing seasonal patterns in precipitation and fire occurrence, correlation of different rainfall accumulation periods to fire occurrence, peat vs. non-peat fires, diurnal fire occurrence, spatial patterns, links to ENSO and IOD), some of which are not treated with the required level of detail.

I would suggest that the authors revise their manuscript, concentrating on the main findings and describing and discussing them in greater overall detail.

Specific comments

Number of fires vs. area burnt. In many cases (e.g. l. 101-102 (Introduction)), the authors refer to the number of fires in a specific area only. Please note that in a peat/vegetation fire context, the area burnt (if available) may be more meaningful or at least a valuable addition to number of fires. In addition, stating the number of fires per unit area may also be helpful for comparing the different areas (e.g. in lines 236-244).

Matching GPCP and TRMM data, dry season definition. Why are only the 4 driest months of the year used for regression (l. 168)? Later it is stated that correlations were low for part of the region due to low variability of the dry month data. Was the regression performed with daily or monthly data? Please report in more detail. Later on, are lines 252-265 referring to the same dry period definition? If no, how was this period defined (daily vs. monthly, consecutive dry days vs. lowest sum of precipitation)?

Combining MODIS and ATSR data. Please add information on how this was achieved in a mathematical sense (linear regression?) How was it possible to compensate for lower ATSR sampling rates (l. 192-203).

Analysis of fire occurrence vs. rainfall thresholds/accumulation periods. In the Introduction, time frames for rainfall accumulation are duly described (l. 207-210). However, the methods for processing these data and combining them with active fires (Results l. 274-293, Fig. 8, Table 3) are not mentioned. More detailed information is needed here as well, as this part of the manuscript is currently not reproducible.

Analysis of diurnal fire occurrence. This aspect is currently present but not well covered in the manuscript, i.e. the main analysis is focusing on night-time data, there are few references to other studies of diurnal fire activity and the sections dealing with this are

rather weak. Additional 10:30 AM data from Terra would have been available but were not used and no attempt for a comparison to meteorological data (which would have to be diurnal as well) was reported. Consider either expanding or removing this aspect.

Links to ENSO and IOD. This is coming up somewhat surprisingly when Fig. 9 is first mentioned in l. 310 (Discussion) although it would be an interesting aspect if covered in more detail and in all appropriate sections of the manuscript.

Technical corrections

l. 53 and onwards: add a space between values and units (e.g. 117,000 km$^2$)

l. 54 GFED4s This abbreviation has been explained in the abstract; please do so in the text as well.

l. 55 20kgC per m-2, change to: 20 kg C m-2

l. 61 represents, change to represent

l. 70-72 Is it possible to cite a reference confirming this?

l. 91-92 move 'region' in front of the brackets

l. 93 insert a comma after the 2nd citation

l. 98 change 'then' to 'at the time' or similar

l. 98 add 'However, . . .' before 'Fires do not solely occur. . .'

l. 99-100 Giglio (2010) is missing in the reference list

l. 117 Please clarify that you mean extinction of light and not extinction of fires.

l. 123-124 In this context, 1997-98 is not a fire event, but rather a fire season. Also applies to fire years.

l. 134 and further occurrences, also for tables: check if 'fig X' is the correct way to refer to figures.

l. 160-161 I would guess that 0.25° is the final product spatial resolution and not the rain gauge resolution. Please rephrase to clarify this.

l. 170-173 Here you repeatedly report the results of a linear regression as 'correlation'. The correct term would be coefficient of determination.

l. 180 amount of fires: change to number of fires?

l. 185 change comma to semicolon or to 'where'

l. 186 2x 'use' in one sentence

l. 193-195 and elsewhere MODIS data is sometimes called MODIS, MODIS Terra or just Terra. Please harmonize; also applies to Aqua mentioned later on.

l. 209 Build-up of rain sounds inappropriate; how about accumulated precipitation, sum of rain or similar?

l. 215 Wetlands International (2015) is missing in reference list.

l. 220 and 223 Refer to parts a) and b) of Fig. 4 or include the area names in the figure for easier reference.

l. 230-231 Move the naming conventions to the Methods section.

l. 273 A reference to Fig. 7 is made, but as far as I understand Fig. 7 is not showing the data discussed here (120 driest days vs. monthly rainfall).

l. 274 Please add a comma after 'all regions studied'.

l. 310-311 The properties of the two El Niño years could also be mentioned in the introduction.

l. 322-328 Are you sure the years and the reference to Fig. 6 are correct? Years 2005 and 2014 are not shown there and neither is northern Sumatra. If you meant to refer to Fig. 5, the 'monthly fires' line is much too compressed to observe any of the details reported here.

l. 337 Do you mean Field and Shen (2008)?

l. 372-373 How do you define the start of the fire season and is it not related to the meteorological preconditions?

l. 376 Consider rephrasing to: '... there were more days with rainfall in 2015 than in 1997.' or similar.

References: some references (usually with URL addresses) have a different formatting/font size (e.g. l. 421) and sometimes the doi is reported differently than for most other references (e.g. l. 446-447). There are some unusual symbols in my PDF version (l. 428 and 467), a comma is missing between doi and year in l. 462 and there is a blank in 'Nino' in l. 532.

Tables 1-3 Please explain the area codes in the table caption or spell them out in the table.

Fig. 1 State what 'active fires' (number of fires detected?) means. What is the white area?

Fig. 2+3 Consider merging those two figures.

Fig. 4 Please consider writing the area name after a), b) etc. in each panel for easier reference. Also, consider if adding a legend and monthly mean temperatures may be helpful for the reader.

Fig. 5 The monthly fire detections are hardly visible in this figure. If this information is vital, consider a different arrangement of the panels or a different plotting technique. Please also add the area names directly in the panels.

Fig. 6 Data from 2006 does not seem to be mentioned in the manuscript. Is this intentional?

Fig. 7 Do you mean monthly sum of precipitation? – Creating averages would hardly be useful for this parameter.

Fig. 8 Please consider reporting the years as 99, 00, 01 etc., if possible.

Fig. 9 What is the data source for this figure? How is 'strongest fire activity' defined? Would it be useful to add sea surface temperature of the El Niño 3.4 region and to show the years 1997 and 2015 that are discussed in the text in greater detail?

————————————————

---

## Referee Comment (RC2) · D. C. Zemp (Referee) · 14 Dec 2016

General comments The authors examine that the relation between rainfall and fire in Indonesia, with a particular focus on the regions with high number of fire events. Their main findings are of interest for the community of Earth System scientists and beyond, in particular those regarding thresholds of accumulated rainfall prior to high fire years, as well as the importance of minor rain events during the dry season to limit fires, as this has potential implication for predictions and model development. The results are generally clearly presented and discussed by referring to relevant previous studies. My major concerns relate to the methodology based on linear correlations for the merging of the datasets. In my point of view, this approach is over-simple as it omits potential non-linearity in the different datasets. This is problematic as the authors use the merged time series to conclude on the differences in fire and rainfall patterns during

1997 and 2015. Furthermore, as it is clearly mentioned in the introduction, fires in Indonesia are tightly coupled with land-use change. The Mega Rice project is mentioned in the discussion of the results, but it would be interesting to include land-use change data in the analysis of spatio-temporal variability of fires as well. If the authors think that it is outside the scope of the study, they should in my opinion tune down the conclusions on the role of rainfall patters to explain fire events.

Specific comments

1) Description of the study region (L. 135-143) This section uses a lot of vague expressions such as "most of", "largely covered by", "abundant", "a large scar of", "widespread area", "extensively"... The authors should give quantitative estimates and provide associated references.

2) Merging precipitation datasets To merge the precipitation datasets, the authors use a linear regression based on the 4 driest months (L. 168). What is the rationale behind this procedure? Unless the authors have a solid justification for this choice, they should deepen the analysis of the (linear or non-linear) similarities between the full time series (and not only dry months). For example, while a linear correlation seems appropriate in most of Indonesia (as shown by high R2 values in Fig. 2b), it is not in the case in the northern part of the country. The authors mention the low inter-annual variability as explanation, but this statement should be justified by quantitative analyses. To strengthen the methodology, further assessment of a possible non-linearity between both time series would be appreciated.

3) Merging fire datasets: The authors mention the spatial resolution of the Terra product (L. 182), but not of ATSR. Please give this information. Why are the data combined at 1° lon/lat grid? Is this the resolution of ATSR? The "correction factor" that the authors calculate (L. 195) again assumes linear correlation between the time series without justification. The authors show this correction factor for grid cells with high fire detections (Fig. 3b), but in my point of view they should deepen the analysis of non-linear

similarities between the time series.

"Our fire correction factor compensated for the lower sampling rates from ATSR over Terra between 2001 and 2011 in all Indonesia (R2 is 0.97)" (l. 200). The authors should better explain their arguments. What correlation was quantified?

How is the "fraction of daytime fire burning over night" (l. 211) calculated? Is this ratio calculated on a yearly or daily basis? The authors should give more details on their methods.

3) Results

Fig. 1: What is the unit: number of active fires per year? Furthermore, I suggest to increase the size of the colorbar and to add ticks in order to make the figure more informative.

Fig. 5: It is not clear what is meant by "monthly fires" and "annual fires" in the figure caption.

Fig. 9: Is it an average over Indonesia? Perhaps this could be specified.

—————————————————————

---

## Author Comment (AC1) · 3 Feb 2017

Dear reviewer
We appreciate the comments on our draft. Please find a detailed response below with the reviewer's comments in italics.

Regards,

Thierry Fanin

[Figure]

**1 Specific comments**

1. *Number of fires vs. area burnt. In many cases (e.g. l. 101-102 (Introduction)), the authors refer to the number of fires in a specific area only. Please note that in a peat/vegetation fire context, the area burnt (if available) may be more meaningful or at least a valuable addition to number of fires. In addition, stating the number of fires per unit area may also be helpful for comparing the different areas (e.g. in lines 236-244).*

   We agree with the reviewer. In general, active fires are to some degree the product of burned area and fuel consumption and thus more closely related to emissions than burned area. However, burned area observations may be more successful in peatlands if the emissions are too small to be detected by active fires. Given that the larger peat fires are in general detected by active fire observations and because from ATSR only active fire observations are available we focused on active fire detections. We now more clearly state this line of reasoning in the abstract.

2. *Matching GPCP and TRMM data, dry season definition. Why are only the 4 driest months of the year used for regression (l. 168)? Later it is stated that correlations were low for part of the region due to low variability of the dry month data. Was the regression performed with daily or monthly data? Please report in more detail. Later on, are lines 252-265 referring to the same dry period definition? If no, how was this period defined (daily vs. monthly, consecutive dry days vs. lowest sum of precipitation)?*

   We used the 4 driest months because of the different regional dry seasons (see Figure 4). We tried other time frames, from 1 to 5 months but a 4 month window yielded the highest correlations. We performed the regression using monthly data

which is now more clearly stated in the methodology. Regarding the discussion part you are referring to, we used the 120 driest days based on daily data. The reason we choose a different approach is that to merge GPCP and TRMM, we needed a yearly consistent time period, while in this later discussion section we focus on specific years for which we can better use daily data. To avoid confusion, we followed you advice and modified the definition in the manuscript.

3. *Combining MODIS and ATSR data. Please add information on how this was achieved in a mathematical sense (linear regression?) How was it possible to compensate for lower ATSR sampling rates (l. 192-203).*

To compensate the lower ATSR sampling rate we multiplied the ATSR data with a monthly correction factor. To calculate this monthly correction factor we used the monthly total ATSR and MODIS active fire counts between 2001 and 2012. We then calculated the ratio at 1 degree spatial resolution, dividing MODIS by ATSR. This correction factor accounts for differences in sampling rates.

4. *Analysis of fire occurrence vs. rainfall thresholds/accumulation periods. In the Introduction, time frames for rainfall accumulation are duly described (l. 207-210). However, the methods for processing these data and combining them with active fires (Results l. 274-293, Fig. 8, Table 3) are not mentioned. More detailed information is needed here as well, as this part of the manuscript is currently not reproducible.*

We now more clearly state in the methodology how we combined active fire and rainfall (l. 225-226). We mentioned that we calculated the rainfall accumulation in each grid cell prior each active fire detected on time frames of 0, 7, 15, 30, 60, 90 and 120 days.

5. *Analysis of diurnal fire occurrence. This aspect is currently present but not*

*well covered in the manuscript, i.e. the main analysis is focusing on night-time data, there are few references to other studies of diurnal fire activity and the sections dealing with this are rather weak. Additional 10:30 AM data from Terra would have been available but were not used and no attempt for a comparison to meteorological data (which would have to be diurnal as well) was reported. Consider either expanding or removing this aspect.*

To achieve our relatively long time series we had to limit ourselves to night-time data simply because ATSR has only night-time data. If the diurnal cycle has changed over our time period this would have had an impact on the results. We prefer therefore to keep the discussion about the diurnal cycle to make the reader aware of this caveat.

6. *Links to ENSO and IOD. This is coming up somewhat surprisingly when Fig. 9 is first mentioned in l. 310 (Discussion) although it would be an interesting aspect if covered in more detail and in all appropriate sections of the manuscript.*

ENSO and IOD are explained in the introduction and then throughout all the discussion and conclusion. To answer the referee's comment, we now mention the use of ENSO and IOD data in the methodology section, citing where we collected the data (l. 235-236). We also mention the link between El Niño and the amount of active fires in the results section (l 246 – 253)

**2 Technical corrections**

1. *l. 53 and onwards: add a space between values and units (e.g. 117,000 km2)*
Corrected

2. *l. 54 GFED4s This abbreviation has been explained in the abstract; please do so in the text as well.*
Corrected

3. *l. 55 20kgC per m-2, change to: 20 kg C m-2*
Corrected throughout the manuscript

4. *l. 61 represents, change to represent*
Corrected

5. *l. 70-72 Is it possible to cite a reference confirming this?*
Done

6. *l. 91-92 move 'region' in front of the brackets*
Corrected

7. *l. 93 insert a comma after the 2nd citation*
Corrected

8. *l. 98 change 'then' to 'at the time' or similar*
Corrected

9. *l. 98 add 'However, : : :' before 'Fires do not solely occur: : :'*
Corrected

10. *l. 99-100 Giglio (2010) is missing in the reference list*
Corrected

11. *l. 117 Please clarify that you mean extinction of light and not extinction of fires.*
Corrected

12. *l. 123-124 In this context, 1997-98 is not a fire event, but rather a fire season. Also applies to fire years.*
Corrected

13. *l. 134 and further occurrences, also for tables: check if 'fig X' is the correct way to refer to figures.*
We modified it to "figure". The abbreviation "fig" must only be used if it appears in running text.

14. *l. 160-161 I would guess that* $0.25_i sthefinalproductspatialresolutionandnottheraingaugeresolution. Pleaserephrase$
$Corrected$

15. *l. 170-173 Here you repeatedly report the results of a linear regression as 'correlation'. The correct term would be coefficient of determination.*
Corrected throughout the manuscript

16. *l. 180 amount of fires: change to number of fires?*
Corrected

17. *l. 185 change comma to semicolon or to 'where'*
Corrected

18. *l. 186 2x 'use' in one sentence*
Corrected

19. *l. 193-195 and elsewhere MODIS data is sometimes called MODIS, MODIS Terra or just Terra. Please harmonize; also applies to Aqua mentioned later on.*
Corrected

20. *l. 209 Build-up of rain sounds inappropriate; how about accumulated precipitation, sum of rain or similar?*
Corrected

21. *l. 215 Wetlands International (2015) is missing in reference list.*
Corrected

22. *l. 220 and 223 Refer to parts a) and b) of Fig. 4 or include the area names in the figure for easier reference.*
Corrected

23. *l. 230-231 Move the naming conventions to the Methods section.*
Corrected

24. *l. 273 A reference to Fig. 7 is made, but as far as I understand Fig. 7 is not showing the data discussed here (120 driest days vs. monthly rainfall).*
For better consistency between the manuscript and the figure we changed the average results explained previously to cumulated rainfall as shown in the figure.

25. *l. 274 Please add a comma after 'all regions studied'.*
Corrected

26. *l. 310-311 The properties of the two El Niño years could also be mentioned in the introduction.*
We added a sentence in the introduction

27. *l. 322-328 Are you sure the years and the reference to Fig. 6 are correct? Years 2005 and 2014 are not shown there and neither is northern Sumatra. If you meant to refer to Fig. 5, the 'monthly fires' line is much too compressed to observe any of the details reported here.*
Yes the wrong figure was mentioned and corrected to Fig. 5. Following your recommendation, we moved the monthly data to the second y-axis for better clarity.

28. *l. 337 Do you mean Field and Shen (2008)?*
Corrected

29. *l. 372-373 How do you define the start of the fire season and is it not related to the meteorological preconditions?*

We defined fire season as the period with most active fires in each regions, mentioned in the first paragraph in the results section and shown in figure 4.

30. *l. 376 Consider rephrasing to: ': : : there were more days with rainfall in 2015 than in 1997.' or similar.*
    Corrected

31. *References: some references (usually with URL addresses) have a different formatting/ font size (e.g. l. 421) and sometimes the doi is reported differently than for most other references (e.g. l. 446-447). There are some unusual symbols in my PDF version (l. 428 and 467), a comma is missing between doi and year in l. 462 and there is a blank in 'Nino' in l. 532.*
    We modified the doi, but difference in formatting or unusual symbols could not be found on our version.

**3 Figures**

1. *Tables 1-3 Please explain the area codes in the table caption or spell them out in the table.*
   Corrected

2. *Fig. 1 State what 'active fires' (number of fires detected?) means. What is the white area?*
   We corrected the definition of 'active fires' in the figure caption. The white areas are where no active fires have been detected and this is now explained in the caption.

3. *Fig. 2+3 Consider merging those two figures.*
   We decided not to merge these two figures because they represent two different parts of the methodology

4. *Fig. 4 Please consider writing the area name after a), b) etc. in each panel for easier reference. Also, consider if adding a legend and monthly mean temperatures may be helpful for the reader.*
   We modified the area names and modified the colors of the y-axis labels to improve clarity. Finally, we have investigated monthly temperature but this adds very little with regard to explaining variability in fire activity.

5. *Fig. 5 The monthly fire detections are hardly visible in this figure. If this information is vital, consider a different arrangement of the panels or a different plotting technique. Please also add the area names directly in the panels.*
   According to referee comments, we changed the monthly fire detection arrangement and added area names.

6. *Fig. 6 Data from 2006 does not seem to be mentioned in the manuscript. Is this intentional?*
   Following your recommendation we extended the 2006 explanation (l. 268-269). Less text is dedicated to this year because of the lower fire activity that year. Yet, we decided to keep 2006 in this figure to show the reader the difference in fire and rainfall activity between weak and strong El Niño years.

7. *Fig. 7 Do you mean monthly sum of precipitation? – Creating averages would hardly be useful for this parameter.*
   Corrected

8. *Fig. 8 Please consider reporting the years as 99, 00, 01 etc., if possible.*
   We appreciate the suggestion and have tried but unfortunately that makes the figure hard to read. We do state in the caption how the numbers should be interpreted.

9. *Fig. 9 What is the data source for this figure? How is 'strongest fire activity' defined? Would it be useful to add sea surface temperature of the El Niño 3.4*
*region and to show the years 1997 and 2015 that are discussed in the text in greater detail?*

Following referee's advice, we added the data source for both dataset. Strong fire activity was defined as month with more than 5000 AF. Trying different possibility to present ENSO and IOD, we found that showing trends in El Niño or La Nina was more effective in passing the message.

---

## Author Comment (AC2) · 3 Feb 2017

Dear Dr Zemp
We appreciate the comments on our draft. Please find a detailed response below with the reviewer's comments in italics.

Regards,

Thierry Fanin

**1  Specific comments**

1. *Description of the study region (L. 135-143) This section uses a lot of vague expressions such as "most of", "largely covered by", "abundant", "a large scar of", "widespread area", "extensively. . . The authors should give quantitative estimates and provide associated references.*

   We agree with the reviewer and removed these vague expressions and added referenced quantitative estimates. For example we added "Between 1990 and 2010 75.400 km2 of primary forest was cleared (Margono, 2013)" or "The other side of the island with the provinces of Riau, Jambi, and South Sumatra is largely covered with rain fed agriculture for subsistence and commercial forestry with agricultural activities (114.928 km2 and 44.307 km2 respectively, (LADA, 2008))"

2. *Merging precipitation datasets To merge the precipitation datasets, the authors use a linear regression based on the 4 driest months (L. 168). What is the rationale behind this procedure? Unless the authors have a solid justification for this choice, they should deepen the analysis of the (linear or non-linear) similarities between the full time series (and not only dry months). For example, while a linear correlation seems appropriate in most of Indonesia (as shown by high R2 values in Fig. 2b), it is not in the case in the northern part of the country. The authors mention the low inter-annual variability as explanation, but this statement should be justified by quantitative analyses. To strengthen the methodology, further assessment of a possible non-linearity between*

   We have tested several regression types but linear relations gave the highest correlations and also fundamentally most defendable. In addition, we aimed to keep our approach consistent for all of Indonesia. We agree that in some of the northern regions the regression was poor but as stated in the text these regions

contributed very little to total fire detections. Over all Indonesia, 93

3. *Merging fire datasets: The authors mention the spatial res-*
   *olution of the Terra product (L. 182), but not of ATSR.*
   *Please give this information. Why are the data combined at*
   *$1_ion/latgrid? Is this the resolution of ATSR? The "correction factor" that the authors calcu$*
   *$linear similarities between the time series.$*

   The resolution chosen is somewhat arbitrary but we aimed to balance having
   enough spatial detail while keeping the impact of different sampling strategies
   between MODIS and ATSR (and the low number of observations from the latter)
   minimal. As mentioned above, we have tried different regression types (e.g.,.
   linear, exponential) with less success, mainly in grid cells with lower fire activity (l
   206-207). Overall, we feel that the high coefficient of determination (R2 of 0.97)
   between ATSR and MODIS justified this approach and have added additional
   text (l 212-213) where we discuss the spatial variability in the correction factor,
   mentioning that all grid cells with a high correction factors had a low amount of
   active fires.

4. *"Our fire correction factor compensated for the lower sampling rates from ATSR*
   *over Terra between 2001 and 2011 in all Indonesia (R2 is 0.97)" (l. 200). The*
   *authors should better explain their arguments. What correlation was quantified?*

   We clarified this statement by explaining that we applied this correction factor to
   the ATSR data for the whole time series on a monthly time step. The correlation
   was the coefficient of determination of monthly Terra and corrected ATSR for the
   period where both datasets were available.

5. *How is the "fraction of daytime fire burning over night" (l. 211) calculated? Is this*
   *ratio calculated on a yearly or daily basis? The authors should give more details*

   *on their methods.*

   Following the referee's recommendation we added details on our method in line
   231-233. Specifically, we have inserted "To observe the proportion of daytime
   fire burning overnight we compared the annual day and night fire. Due to data
   availability, this exercise could only be produced since 2003."

**2  Results**

1. *Fig. 1: What is the unit: number of active fires per year? Furthermore, I suggest*
   *to increase the size of the colorbar and to add ticks in order to make the figure*
   *more informative.*
   We modified the caption mentioning that we show the sum of AF between 1997
   and 2015. As recommended by the referee, we increased the colorbar size and
   added ticks.

2. *Fig. 5: It is not clear what is meant by "monthly fires" and "annual fires" in the*
   *figure caption.*
   We have modified it to "number of annual and monthly active fire detections" to
   make it more clear.

3. *Fig. 9: Is it an average over Indonesia? Perhaps this could be specified.*
   The results are the general indices in Niño 3.4 for ENSO and the Indian Ocean
   for IOD as mentioned in the introduction and added now to the caption

**3 References**

LADA. Mapping Land Use Systems at global and regional scales for land Degradation Assessment Analysis, 2008.

Margono, B. A.: Mapping deforestation and forest degradation using Landsat time series: a case of Sumatra—Indonesia, 2013.

---

## Author Response (AR2)

Dear Dr Rammig,
Please find the detailed answer to the reviewers, followed by the revised manuscript.

**Reviewer 1:**

**Specific comments**

*Both the manuscript and scientific community would benefit from a further extension of the sections regarding merging fire and precipitation datasets (more detailed explanations including alternatives tried but not chosen).*
In order to better inform the reader about the precipitation merging methods we have included the results of our time frame tests (l. 178-180). In lines 218 we explained that we could have used GFED burned area but instead opted for using only night-time MODIS active fire observations to ensure compatibility with ATSR. In fact, one of our key findings is that this led to a different ratio between the years 1997 and 2015 when comparing our work with GFED. Finally, the decision to use the ATSR-MODIS and GPCP-TRMM datasets was largely predicated on the ability to merge the datasets and their temporal availability. ATSR- MODIS were mainly chosen due to their similar acquisition times at 10.30pm (l .186-194). GPCP-TRMM was chosen to improve spatial resolution. GPCP was available for the entire study period but at only 1 degree spatial resolution, making it difficult to observe spatio-temporal variances at regional scale. Using TRMM combined with the 1997 GPCP allowed us to improve the spatial resolution to 0.25 degree (160-171). This comment was not fully clear to us and we have tried to answer it as completely as possible according to our understanding.

*Although there is now a brief sentence on the use of ENSO and IOD in methods, Fig. 9 is still not mentioned in results and appears only in the discussion.*
We included a short sentence in the result section (l. 326). We did not include extensive discussion of Figure 9 in the results because these results do not originate from our study. The purpose of this figure is more descriptive and reminds the reader about the strong El Niño events of the past two decades.

**Technical corrections**

*line 26 fierce: consider a more objective expression*
We changed 'fierce' to 'severe'

*line 56 change kg C per m-2 to kg C m-2*
corrected

*line 215 consider revising Indonesia en the resulting R2*
corrected

*line 227 what the accumulated rainfall for rehprase dropping 'what'*
corrected

*line 238 sea temperature fluxes Are these really fluxes or merely fluctuations?*
corrected

*line 294 Northern Sumatra had rainfall accumulation insert 'a'*
corrected

**Reviewer 2:**

*I thank the authors for their response to my specific comments. However, they did not address my general comments, in particular my suggestion to include land-use change in the analysis or, if not, to tune down the conclusion of the paper.*

We did not include land use change because of a lack of available datasets during the entire timeframe of our study. This explanation is now included in the "dataset and method" section (l. 240-243). Furthermore, we do provide more anecdotal information, for example regarding the influence of the mega rice project on the 1997 fires. We now not only mention it in the discussion (l. 344-346) but also in the conclusion (l. 399-400). By adding this section in the conclusion, it tunes down our findings by informing the reader that precipitation is not the only driver and we highlight the role of land use change, but we also note that this is not included in our analysis

*The authors write "We also tested whether different time frames gave different results and ended up with the 4-month time window." (L. 269-270) but do not show additional results to support their statements. I think the authors should improve transparency in the description of their methodology and better justify their statements.*

For better transparency in our methodology, we have included the results arising from the different time frames that were tested. These are reflected, along with the chosen timeframe, in Table 1 (l. 178-180). You will notice that the chosen time frame does not significantly impact the result.

[revised manuscript text omitted]

---

## Author Response (AR3)

Dear Dr. Zemp

We appreciate the comments on our draft. Please find a detailed response below with the reviewer's comments in italics.

**Specific comments**
*The authors did still not address my concern properly. Worse, by trying to justify their choices and explain their methodology, they create confusion and mistakes: "We used a 4-month time period because this yielded the highest correlation (R2 = 0.69)." (L. 177). However, from table 1*

*I see that R2 decreases with the length of the time window, so that in fact the 4-month doesn't lead the highest R2. The reason for this decrease is not properly discussed. They continue "However, the results are relatively insensitive to this time period, lowering or lengthening the time period gave comparable results (Table 1)". In my opinion, it makes no sense to justify this statement by showing R2 for different time windows. They should rather use this table to explain*

*and discuss their approach from a statistical point of view, as mentioned above. Instead, the authors should have performed a sensitivity analysis to see how this actually affect the results (ex Fig.2).*

The reviewer correctly notes that our choice for the 4-month time window was not described well. We have redone all analyses using different time windows but in the end settled again on a 4-month window and show the sensitivity to using different time windows. First of all, the $R^2$ values are very similar (Figure R1) for the time frames between 2 and 8 months. While in principle it might make most sense to chose 2 months (which had the highest $R^2$) we prefer to keep 4 months because, once applied to GPCP, the fit between adjusted GPCP and TRMM is best for 3 and 4 months for grid cell with fires (Figure R1, 3 months yields a slope of 1.01 and an $R^2$ of 0.97, and 4 months has a slope of 0.99 and an R2 of 0.97). This is now more clearly justified in the manuscript with this paragraph:
"We investigated time frames from 2 to 8 months. Our choice for 4 months was to some degree arbitrary because the differences are very small when looking at the fit between converted GPCP

and TRMM. However, converted GPCP based on a 4 month time frame to calculate the correction factor yielded close resemblance for the overlapping period with TRMM in grid cells where a fire occurred during our study period (slope: 0.99, R2: 0.97). Results are relatively insensitive to the choice of the time periods as will be shown in the result section. "
In the manuscript we also added a paragraph in our results to demonstrate the sensitivity of the different time frames to calculate the correction factor. In 1997 the mean rainfall for Indonesia varied between 51.7 and 52.9 mm during the dry season (Fig. 10) depending on the time frame chosen. This indicates that our new rainfall dataset is not very sensitive to the choice of time frame. Furthermore, we have tested the different time frames on the experiment from Figure 9. We show that with a 120 days buildup in 1997, rainfall accumulation before a fire would have varied between 239 and 248 mm (Fig. 11). This shows that the choice of time frame does not influence the key findings from this study. The strong non-linear response of fire to rainfall would not change, as 1997 is the only year affected by the choice of correction factor, the remaining years being based on TRMM data.

[Figure]

Figure R1: Spatial distribution of slope (left panels) and coefficient of determination (R², panels in the middle) from a linear regression between GPCP and TRMM rainfall based on the total rainfall for the 2 to 8 driest months between 1998 and 2014. The scatter plots in the right panels show the corrected monthly GPCP and TRMM
rainfall rates for the same time period in grid cells where a fire occurred between 1997 and 2015.

*In addition, as mentioned in my first review report, the statistical analysis that the authors perform for the merging of the fire dataset is also weak (for example, they claim to have tested several regressions but no results appear in the manuscript).*

We made a new figure (Figure R2) showing the results of the 3 different methods tried (ratio, linear regression, exponential regression). We decided not to include it in the manuscript to avoid confusion for the reader. With this Figure we demonstrate that the ratio was the most appropriate method to combine our fire datasets. Furthermore, to make a monthly regression method, we needed several points (monthly fires) in each grid cell. Unfortunately some grid cells have only one fire in a particular month during our study period, resulting in omitting to correct these fires. For example, if between 2001 and 2011 a 1 quarter-degree cell had only one year with fires in November, this particular fires will not be corrected as the regression could no be calculated, resulting in an underestimation of regions with lower active fire detections.

[Figure]

Figure R2: Examples of three approaches for adjusting the 1997-2000 ATSR data. All values are in number of fires. The first column shows the ratio method used in the paper, the second a linear fit, and the third one and exponential fit. The top row shows the number of September fires in a 1° grid cell (2.5°S, 114.7°E, east of Palangkaray, southern Kalimantan) between 2001 and 2011. The ratio method uses the total number of MODIS and ATSR in a particular month (here: September) for those 11 years to calculate the conversion factor (ratio). For the linear and exponential regression method the number of yearly fires in a particular month is used to calculate the regression that would then be used as a conversion factor. Not all grid cells have fires in each month, resulting in poorer linear or exponential fit. All approaches vary spatially as shown in Figure 3 with the ratio method. The bottom row shows how the adjusted number of ATSR active fire detections compares to the MODIS data when the different conversion approaches are used for the whole domain.

[revised manuscript text omitted]